# Over four minutes of pyruvate $T_1$ using chemically and physically induced deceleration of relaxation

Josh P. Peters [1] ✉, Florin Teleanu [2,3], Huijing Zou[2], Ehtisham Rasool[1], Farhad Haj Mohamad [1], Heiner Schäfer [4], Jan-Bernd Hövener [1], Alexej Jerschow[2] & Andrey N. Pravdivtsev [1] ✉

[1-$^{13}$C]pyruvate is the most widely used probe for hyperpolarized metabolic magnetic resonance imaging, with profound applications in tumor and inflammation diagnosis as well as treatment monitoring. The most fundamental hurdle to broader application, however, remains the rapid relaxation of polarization and the associated signal loss. Here, we report a method to address this challenge. Studying the nuclear spin relaxation dispersion of [1-$^{13}$C]pyruvate across magnetic fields from 8 µT to 9.4 T, as a function of additives, solvents, and preparation methods, allows us to achieve $T_1$ relaxation times of up to four minutes. Such a long $T_1$ could enable reliable quality control and nearly polarization loss-free transport, further boosting the power of hyperpolarized metabolic MRI. The power of the optimized protocol for $T_1$ elongation in vitro is showcased by more than doubling the sensitivity in a metabolic experiment.

The usefulness of hyperpolarized metabolic magnetic resonance[1,2] hinges on the relation of two periods of time: the time required for the agent to trace the desired process or function in vivo (metabolic activity), and the lifetime of the hyperpolarization (observable time). The first period is primarily determined by physiology, whereas the second period must allow time for quality control, sample transfer, administration, in vivo distribution, and tracing the desired function, without the polarization decaying below the detection threshold.

Unfortunately, many long-lived probes, such as $^{15}$N-labeled molecules with chemically and magnetically isolated nitrogen, or certain $^{13}$C molecules[3], do not undergo relevant metabolic transformations, limiting their utility for studying physiological processes like perfusion[4–6]. Others, such as $^{13}$C-labeled glucose, undergo rapid metabolism but exhibit equally rapid relaxation, thereby impeding human application[7]. Some $^{15}$N molecules have a long relaxation time at high fields but a short one at low fields[8]. [1-$^{13}$C]pyruvate, however, features sufficiently fast metabolism and comparatively slow relaxation and thus has become the most widely used hyperpolarization probe[1,9]. Even for pyruvate, however, many applications are limited by the signal-to-noise ratio, which affects spatial and temporal resolution and sensitivity.

The key factor for MRI is the polarization available at the time of detection. This factor is determined by (a) the initial polarization and (b) the loss between polarization and detection. Many efforts have been made to maximize the initial polarization, and to date, about 50% or more of the theoretical in general or for specific agents?[10]. Reducing the loss between polarization and detection is equally important but has received less attention yet (e.g., by prolonging $T_1$[11] or by polarizing in the MRI[12,13]).

The currently most prominent hyperpolarization methodology is dissolution dynamic nuclear polarization (dDNP)[14], with many clinical examples to date[1,15]. Parahydrogen-induced polarization (PHIP)[16,17] and signal amplification by reversible exchange (SABRE)[18,19] are rapidly developing viable alternatives for liquid-state hyperpolarization.

[1]Section Biomedical Imaging (SBMI), Molecular Imaging North Competence Center (MOINCC), Department of Radiology and Neuroradiology, University Hospital Schleswig-Holstein, Kiel University, Kiel, Germany. [2]Department of Chemistry, New York University, New York, NY, US. [3]ELI-NP, "Horia Hulubei" National Institute for Physics and Nuclear Engineering, Bucharest-Magurele, Ilfov, Romania. [4]Institute for Experimental Cancer Research, University Hospital Schleswig-Holstein, Kiel University, Kiel, Germany. ✉e-mail: josh.peters@rad.uni-kiel.de; andrey.pravdivtsev@rad.uni-kiel.de

Pyruvate is arguably one of the most frequently used hyperpolarization agents[10,20]. Intriguingly, the relaxation lifetime of pyruvate was reported to differ across hyperpolarization methods[21]; however, the sample composition varied slightly. Since the polarization method should not affect the relaxation time, we investigated how the sample composition affects the observed relaxation time of pyruvate. We identified the optimal composition and tested it in vitro following dDNP hyperpolarization. The detailed challenges, comparisons, and state of the art of the hyperpolarization method have been reviewed previously[22–25]. Each hyperpolarization method has its challenges, such as filtration of paramagnetic impurities in dDNP or of metal-organic catalysts or solvents in PHIP/SABRE. The ultimately prepared and purified sample should have the same hyperpolarization lifetime.

Many sample alterations which are discussed in this manuscript have been explored individually in the scope of relaxation studies and hyperpolarization experiments before: chelation agent (EDTA), buffer (Tris), solvent deuteration ($D_2O$), sample degassing (no $O_2$), pyruvate deuteration, and the applied magnetic field (Fig. 1)[8,11,26–31]. For example, previously measured high-field $T_1$ of [1-13C]pyruvate did not reveal a significant impact of pyruvate deuteration on its $T_1$[28], despite the well-known fact that deuteration positively impacts the relaxation properties of remaining spins[29,30]. As shown below, the achieved relaxation value results from a compounding effect of the systematic deactivation of relaxation sources. The effect of chemically induced deceleration of nuclear relaxation (CIDER) was recently discovered, demonstrating that hydrogen-bonding additives can reduce relaxation induced by rapid chemical exchange of quickly relaxing species or by increasing the distance to paramagnetic impurities; it likely plays a key role in explaining the impact of Tris buffer[11]. Tris and EDTA are commonly used in clinical dDNP experiments[20,32] due to their biocompatibility. In addition, $D_2O$ has recently been shown to be safe for human imaging[31,33]. The use of $D_2O$ prolongs $T_1$ relaxation times and preserves polarization levels, thereby increasing the signal-to-noise ratio (SNR) of pyruvate and its metabolite lactate in humans[31].

To probe relaxation at lower fields where magnetization is low, the magnetic field cycling (MFC) concept was previously applied to measure nuclear magnetic relaxation dispersion (NMRD)[34] and estimate polarization losses for hyperpolarized pyruvate[35–37], however, detailed relaxation studies remain scarce.

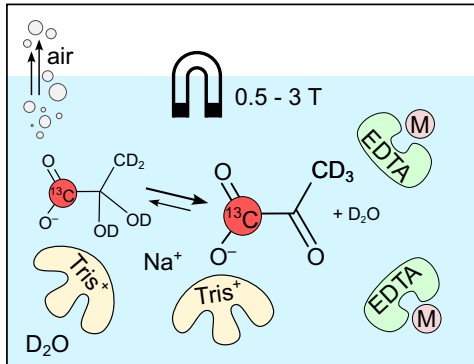

**Fig. 1 | Schematic view of [1-13C]pyruvate and factors leading to more than 4 min 1-13C relaxation time.** The relaxation of pyruvate is affected by intra- and intermolecular spin-spin interactions with nuclear spins and paramagnetic impurities such as $O_2$, chemical exchange, magnetic field, and others. Our investigations identified that the addition of Tris and EDTA, even in pure water, is beneficial for $T_1$. Degassing the sample, solvent deuteration, and deuteration of the methyl group further prolong the relaxation time. EDTA chelates dissolved ions, like M = $Fe^{2+}$, $Mn^{2+}$. Tris could weakly coordinate with its positive charge to pyruvate, shielding it from exchange and impurities through the CIDER effect[11]. All experiments here were performed at a neutral pH of 7.6 ± 0.1.

In this work, we systematically investigate the sample composition of [1-13C]pyruvate and identify conditions under which the relaxation time exceeds 4 min. We assess contributions from various interactions to the relaxation of [1-13C]pyruvate by combining NMRD measurements under different conditions, molecular dynamics simulations, and ab initio computations. An optimized sample is applied to an in vitro cancer model, demonstrating more than a doubling of metabolic sensitivity under optimized conditions compared to normal conditions. The ideal conditions for long-lasting $T_1$ are illustrated in Fig. 1.

## Results and discussion

The "Results and discussion" are organized into three parts. First, we introduce the methodology used to measure pyruvate relaxation as a function of magnetic field and sample composition. Second, we quantitatively dissect the individual relaxation contributions and demonstrate how controlled variations in sample composition enabled their identification and mitigation. This analysis is supported by molecular dynamics and DFT simulations of relevant relaxation parameters. Finally, we validate the optimized sample composition in vitro by demonstrating improved metabolic sensitivity resulting from prolonged relaxation times. Data in this manuscript are presented as mean value ± standard deviation.

### Magnetic field cycling experiments

To understand the impact of sample composition and magnetic field on the relaxation properties of [1-13C]pyruvate, we measured relaxation at thermal equilibrium (i.e., without hyperpolarization) using an MFC system. To accelerate (NMRD) measurements, we first transferred polarization from fast-relaxing $^1H$ (if not deuterated) to slower-relaxing [1-13C] of pyruvate using INEPT[35,38] with refocusing at high field, yielding net magnetization on $^{13}C$. Then, the sample was transferred to the desired field, relaxed for a variable time, followed by returning to the high field for observation.

We varied sample composition, as discussed below, and acquired more than 4300 free induction decay (FID) transients to systematically probe the nuclear relaxation across a wide range of well-controlled parameters (Fig. 2). The previously developed, highly automated MFC system[35] was therefore essential, performing approximately 8600 back-and-forth sample shuttling operations during data acquisition. This experimental series yielded NMRDs at nine different conditions, as explained in Fig. 2. $T_1$ measurements at $B_0 = 9.4$ T were done without shuttling.

When pyruvic acid was dissolved in a double distilled water and adjusted with NaOH to neutral pH, 1-13C relaxation was fastest at low fields (≈8 μT–8 mT, $T_1 = 30.9 ± 0.9$ s), slowest at intermediate fields (≈0.1–1 T, $T_1 = 48.6 ± 2.5$ s), and accelerating again at high fields (≈1–9.4 T, Fig. 2, sample #I). The decrease of $T_1$ at low field is associated with paramagnetic impurities and chemical exchange, while that at high magnetic fields is related to chemical shift anisotropy[39].

Substitution of $H_2O$ (Fig. 2, sample #I as given by the # in panel 2B) with $D_2O$ (sample #H) increased $T_1$ uniformly across the entire field range by ≈15 s. This effect is consistent with intermolecular dipolar relaxation being largely field-independent. The replacement of $^1H$ with $^2H$ reduced the water-induced dipole-dipole relaxation contribution by a factor of $(I_H(I_H + 1))/(I_D(I_D + 1))(\gamma_H/\gamma_D)^2 \approx 15.9$ (eq 11, ref. 40), where $I_H = \frac{1}{2}$ and $I_D = 1$ are nuclear spin values and $\gamma$ corresponds to the gyromagnetic ratio of H and D. Nevertheless, sample #H still showed the shortest $T_1$ of all $D_2O$-containing samples at low fields (43.4 ± 1.5 s).

The addition of Tris buffer to $H_2O$, in conjunction with NaOH for pH adjustment, further increased $T_1$ across the entire field range, with the strongest effect observed at low fields (compare Fig. 2, sample #F and #I). The lifetime was 50.0 ± 0.9 s at low field and reached 59.3 s in the intermediate field range at 2 T. We attribute the effect to Tris chelating paramagnetic species or coordinating with pyruvate through

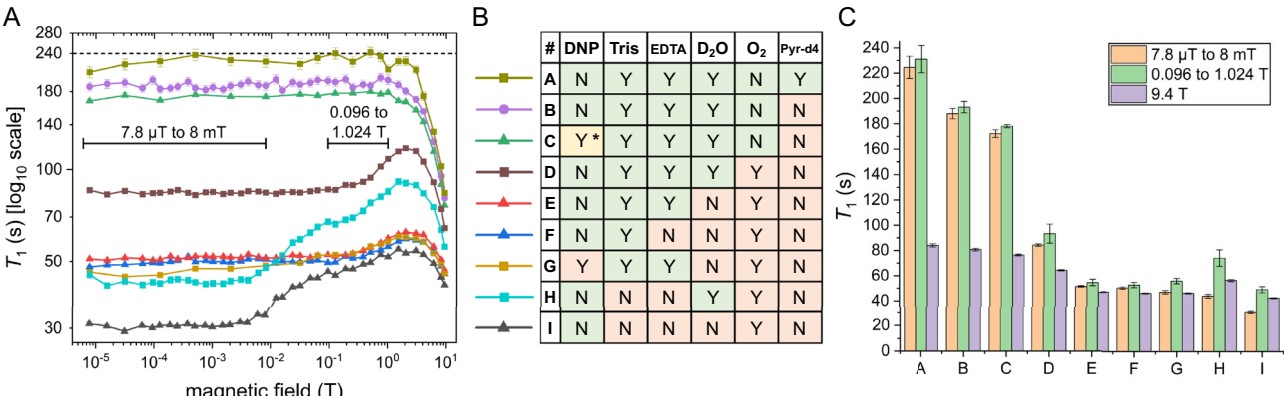

**Fig. 2 | Nuclear magnetic relaxation dispersion (NMRD) of [1–¹³C]pyruvate at fields from 7.8 μT to 9.4 T with different sample compositions.** Addition of Tris, and EDTA, the use of deuterated solvent and pyruvate, and sample degassing boosted $T_1$ of [1–¹³C]pyruvate from about 30 to over 240 s at low magnetic fields. **A** NMRD profiles—$T_1$ (log scale) as a function of magnetic field—for nine sample compositions (#A–#I, colors matching panel (**B**)). Each NMRD has been recorded once, and the error bands present the SD of the fitted $T_1$ values. **B** List of binary composition of each sample: whether DNP radical, Tris buffer, EDTA, D₂O (as solvent), O₂ (degassing status), and perdeuterated pyruvate (Pyr-d4) were included (Y) or excluded (N). Key variables driving $T_1$ enhancement are D₂O substitution, degassing, and addition of Tris/EDTA. **C** Mean $T_1 \pm$ SD for each sample, calculated across the individual field-strength measurements falling within each of the three field ranges highlighted in (**A**) ($n = 6$–18 per range, except $n = 1$ at 9.4 T; see defined

regions in (**A**)). Individual $T_1$ datapoints at each field strength are shown in (**A**) and have not been replotted in (**C**) for visual clarity. The field around 1 T gives the longest relaxation time across all samples, reaching a maximum of $242.1 \pm 9.4$ s at 0.512 T for the sample #A. Details: No INEPT was used on the #A due to the absence of protons in the pyruvate-d₄; therefore, fewer points have been sampled due to increased measurement time (>48 h for the whole NMRD), while other NMRDs were measured with INEPT for acceleration of signal acquisition and signal enhancement. * Indicates that the radical in the dDNP sample has been filtered, and vitamin C was added after dDNP experiment. Abbreviations: DNP dynamic nuclear polarization, Tris tris buffer, EDTA Ethylenediaminetetraacetic acid, Pyr-d4 perdeuterated pyruvate. Exact compositions are given in Supplementary Note 1, Supplementary Table S1. Source data are provided as a Source Data file.

transient ion-pairing and hydrogen-bonding interactions, thereby protecting it from rapidly relaxing species. This effect resembles the chemically induced deceleration of relaxation (CIDER) effect, in which additives shielded probes from fast-relaxing and fast-exchanging nuclei such as protons[11].

To further investigate a possible chelating effect, we compared Tris-buffered solutions with and without EDTA (Fig. 2, sample #E compared to #F). Adding EDTA increased $T_1$ only marginally (1–3 s), suggesting that Tris alone protects against a substantial fraction of the relaxation channels that EDTA targets in otherwise pure H₂O; EDTA is a well-known chelation agent that should protect against metal cations. Combining Tris and EDTA in D₂O in sample #D resulted in a $T_1$ increase of ≈50 s at low fields and ≈20 s at high fields compared to sample #I, or ≈40 s at low fields and ≈10 s at high fields compared to sample #H, which is also a D₂O sample as sample #D but lacks Tris and EDTA, exceeding ≈115 s in the intermediate regime.

Interestingly, the $T_1$ of samples with Tris and EDTA was longer at low field than at high fields, with a maximum in the intermediate regime. This NMRD feature has practical value since relaxation at low and intermediate fields dominates hyperpolarization losses during transfer from the polarizer to the imaging system; prolonging $T_1$ in this range is therefore critical.

The $T_1$ dispersion of [1-¹³C]pyruvate in a standard DNP sample showed relatively fast relaxation at low fields ($46.6 \pm 1.4$ s) and slower at intermediate fields ($55.4 \pm 2.2$ s), followed by a steep acceleration of relaxation at high fields (Fig. 2, sample #G). Still, this sample, although containing trityl radicals, has a longer relaxation time than the pure water sample #I. Radicals, such as Trityl AH111501, are required for the dDNP process as a source of polarization. While chelating agents such as EDTA are typically used to remove ionic impurities from the solution, vitamin C may be used as a redox radical scavenger to lessen the impact of the radicals used for the DNP process[41] (in liquid state, see below).

Nonetheless, the decline in $T_1$ of sample #D from intermediate to low fields indicated contributions from either paramagnetic relaxation or chemical exchange. Degassing the samples via three freeze–thaw

cycles further increased $T_1$, presumably by removing molecular oxygen with the triplet ground state, and eliminated the field-dependent dip at the low-to-intermediate field transition (Fig. 2, sample #B compared to #D). At low field, $T_1$ increased from $84.2 \pm 0.9$ s (sample #D) to $188.0 \pm 4.1$ s (sample #B), indicating that oxygen was the dominant relaxation source at low field after all other sources had been eliminated. Considering the additive nature of the relaxation sources, one can estimate the positive impact from degassing H₂O sample #E: oxygen relaxation contribution at low fields is about 1/84 s⁻¹/ 188 s=0.006 s⁻¹, which would result in an increase of relaxation time for sample #E from 51.5 s to 1/(1/51.5 s–0.006 s⁻¹) ~ 74 s. This 20 s gain is not as impressive as the over 100 s gain in D₂O, indicating that without solvent or substrate deuteration, the over 100 s $T_1$ remains challenging.

Finally, deuteration of pyruvate itself extended $T_1$ at low and intermediate fields by an additional 30–40 s to more than 240 s (compare Fig. 2, sample #A and #B). The average low field lifetime was $224.5 \pm 8.8$ s for sample #A instead of $188.0 \pm 4.1$ s for sample #B, raising the lifetime above 200 s for fields ≤3 T. Pyruvate deuteration has a small effect in high and intermediate fields, explaining why other studies observed only a marginal $T_1$ effect from deuteration[28]— experiments were performed at high fields, where relaxation is much faster and dominated by CSA.

Comparing these insights to the standard pyruvate sample used for metabolic imaging, we find that some conditions match, while others do not. For example, Tris and EDTA are included in the standard sample on an empirical basis, while NaCl is added for physiological isotonicity. When applying all measures suggested above to extend the lifetime (deuteration of the solvent, degassing, vitamin C, radical filtration), the lifetime increased significantly, for example, by more than threefold at low and intermediate fields compared to the standard sample #G, up to $178.1 \pm 1.2$ s for sample #C.

An important methodological difference concerns the degassing procedures. Sample #C was degassed by 3 minutes of argon bubbling, whereas samples #A and #B were degassed using three freeze–pump–thaw cycles. We used argon bubbling for sample #C (and the other dDNP samples discussed below) because the

dissolution medium cannot be maintained under vacuum during loading in DNP, unlike the flame-sealed NMR tubes used for samples #A and #B. Freeze–pump–thaw degassing would leave the solution undersaturated with gas, leading to rapid oxygen uptake upon exposure to air. Argon bubbling removes dissolved oxygen while maintaining gas saturation, making oxygen removal more sustainable and suitable for dissolution media in dDNP, aqueous solution in PHIP, and SABRE experiments. In contrast, freeze–pump–thaw cycles are more appropriate for sealed thermal relaxation studies, as it does not introduce additional substances.

Starting with 100% polarization, and assuming a period of 60 s between polarization and observation that should be sufficient for quality assurance and administration in a clinical setting[31,42], and a magnetic field profile between dDNP system and NMR measured for our settings before (ref. 35, Fig. 6, magnetic profile without transfer magnet), 27.4% will be retained for suboptimal (sample #G) and 70.4% for optimal (sample #C) dDNP samples. Hence, optimizing the sample composition could lead to a 2.6-times increase in SNR. For shorter transfer durations during non-human experiments (such as 19.5 s in ref. 35) the benefit will be lower, with 66% retained for suboptimal (sample #G) and 89% for optimal (sample #C) dDNP samples, therefore SNR is about 1.4-times higher for sample #C.

## Modeling relaxation behavior

By altering the sample's chemical composition, we could isolate individual relaxation contributions and quantify them. This is achieved by subtracting NMRD profiles of sample pairs that differ in a single chemical feature. Then, we fitted the field dependence of these isolated interactions to a corresponding theoretical model and further compared them with predicted rates from molecular dynamics (MD) simulations, generating a detailed picture of the main factors limiting $T_1$ (Fig. 3).

First, we highlight two primary peaks in the $^{13}$C spectrum corresponding to the oxo (Py) and hydrated (PyH) pyruvate forms with a -8.4 ppm chemical shift difference at neutral pH and at room temperature. Under these conditions, their respective chemical shifts are -170.3 ppm (Py) and -178.7 ppm (PyH) with a ratio of $p_{Py} : p_{PyH} \sim 10 : 1$ (see Supplementary Note 6)[43]. Chemical exchange between Py and PyH

is illustrated on Fig. 1. Considering rapid exchange, 0.1–0.2 s$^{-1}$ (Supplementary Note 7 and ref. 44), compared to longitudinal decay rates, 0.015 s$^{-1}$, the observed relaxation rate of the carboxylate $^{13}$C in quantified Py is a weighted-average of two relaxation rates: $R_1^{obs} = p_{Py} R_1^{Py} + p_{PyH} R_1^{PyH}$ (see Supplementary Note 7 for mathematical derivation). This aspect is particularly important in the case of intramolecular dipole-dipole coupling, where the additional hydroxyl protons enhance the relaxation rate of the PyH species and thus reduce the observed $T_1$ of Py as well. At the same time, solvent deuteration will reduce intermolecular dipolar relaxation for both Py and PyH but will also decrease the intramolecular dipolar relaxation rate for PyH, as the hydroxyl protons are replaced with deuterium. Other relaxation interactions, such as CSA and paramagnetic relaxation enhancement (PRE), are assumed to be identical for the two species.

Starting from the chemical composition that maximizes the $^{13}$C polarization lifetime (Fig. 2, sample #A), we notice a dramatic increase in $T_1$ when the magnetic field is lowered from 9.4 T due to the decreasing contribution of the CSA, which scales quadratically with the external field (in the fast motion regime, $R_1^{CSA} = \frac{2}{15} (B_0 \cdot \Delta\sigma \cdot \gamma_{13C})^2 \tau_C^{CSA}$)[45]. This phenomenon impacts all displayed NMRDs, and at fields below 1 T, the CSA contribution is negligible. By fitting the high-field profile and using an MD-derived value for the rotational correlation time of Py at 293 K of $\tau_C^{rot} = 7.3$ ps, we estimate $\Delta\sigma^{exp} \cong 136.5 \pm 1.6$ ppm in good agreement with our DFT calculations of Py in the gas phase ($\Delta\sigma^{DFT} = 133$ ppm) (Supplementary Note 3).

The low field region of the slowest relaxing sample #A, we attributed to a "background" field-independent relaxation, $R_1^0 = 0.0044 \pm 0.0001$ s$^{-1}$, resulting from the additive contributions of the intra- and inter-molecular $^2$H-$^{13}$C dipolar couplings and other remaining contributions from interactions like, possibly, spin-rotation. By running MD simulations of pyruvate in water, we estimated the population-weighted contributions from both $^2$H-$^{13}$C intra- and inter-molecular dipole-dipole interactions (SI), amounting to $R_1^{0, ^2H-^{13}C} = 0.00043 \pm 0.00003$ s$^{-1}$, which represents only 10% of the observed background rate. When comparing deuterated sample #A and protonated #B pyruvate NMRD, we were able to determine the field-independent intra-molecular $^1$H-$^{13}$C dipolar contribution due to

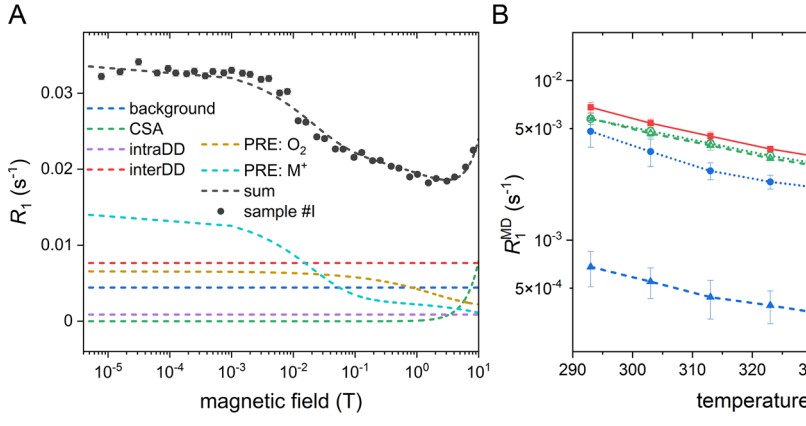

**Fig. 3 | Relaxation contributions to the relaxation of [1–$^{13}$C]pyruvate. A** Field dependence of identified relaxation contributions (dashed lines) adding up to the NMRD profile of [1-$^{13}$C]pyruvate in water (black circles), corresponding to sample #I. Several relaxation mechanisms—inter- and intramolecular dipole-dipole relaxation, PRE from $O_2$ and $M^+$, and CSA—have been isolated by subtracting pairs of NMRD profiles with different chemical compositions and fitting with theoretical equations, resulting in a fit (black line) closely matching the experimental data. **B** Temperature dependence of predicted relaxation rates of longitudinal polarization due to $^1$H-$^{13}$C intra- (blue) or intermolecular (green) dipolar couplings for Py (triangles) and PyH (circles) species as extracted from molecular dynamics simulations for protonated pyruvate and solvent. Due to chemical exchange, the

observable rate for Py was the population-weighted: $R_1^{obs} = p_{Py} R_1^{Py} + p_{PyH} R_1^{PyH}$ (red squares). The temperature dependence of the Py:PyH equilibrium was estimated from experimental measurements (Supplementary Note 6). Overall, dipolar relaxation decreases with increasing temperature. The maximum estimated lifetime of the molecule when only $^1$H-$^{13}$C dipole-dipole relaxation is considered is $2.44 \pm 0.18$ min at 293 K and $6.89 \pm 0.35$ min at 353 K. The data in **A** is from Fig. 2A sample #I. The error bands in **B** present the calculated SD; each datapoint was calculated 10 times. Abbreviations: CSA chemical shift anisotropy, intraDD intramolecular dipole-dipole relaxation, interDD intermolecular dipole-dipole relaxation, PRE paramagnetic relaxation enhancement, PY pyruvate, PYH pyruvate hydrate. Source data are provided as a Source Data file.

the distant methyl protons: $R_1^{\text{intraDD}} = 0.0008 \pm 0.0002\,\text{s}^{-1}$ in good agreement with the MD-predicted value for Py system $R_1^{\text{Py}-\text{intraDD}} = 0.00068 \pm 0.00017\,\text{s}^{-1}$.

Replacing the deuterated water with protonated water adds a field-independent inter-molecular $^1\text{H}$-$^{13}\text{C}$ dipolar contribution to both Py and PyH and an additional intramolecular $^1\text{H}$-$^{13}\text{C}$ dipolar contribution in PyH from hydroxyl protons: $R_1^{\text{inter\&intraDD}}(\text{D}_2\text{O} - \text{H}_2\text{O}) = 0.0076 \pm 0.0002\,\text{s}^{-1}$ (comparing samples #E and #D) and a similar value of $0.0095 \pm 0.0006\,\text{s}^{-1}$ (comparing samples #I and #H). The individual MD-predicted dipolar coupling contributions are shown in Fig. 3B and amount to a population-weighted rate of $0.0062 \pm 0.0005\,\text{s}^{-1}$, where only 6.5% comes from the intramolecular dipolar relaxation due to hydroxyl protons in PyH, such that solvent deuteration enhances $T_1$ by mostly reducing intermolecular interactions.

The dissolved molecular oxygen significantly reduces the $^{13}\text{C}$ polarization lifetime (Fig. 2, sample #D compared to #B) due to its paramagnetic relaxation contribution, $R_1^{\text{PRE:O}_2}$. The maximum value of $T_1$ is approximately reached at 2 T, at an optimal balance between the decreasing CSA and the increasing PRE of $\text{O}_2$. The field-dependence of $R_1^{\text{PRE:O}_2}$ stems from the inter-molecular spectral density for which the tumbling regime is mostly dictated by the electron Larmor frequency and corresponding correlation time[45,46]. We estimate a correlation time of the dipole-dipole electron-carbon interaction of $\tau_{\text{C}}^{\text{PRE:O}_2} = 6.1 \pm 2$ ps (given by oxygen's $T_{1,\text{e}}$) and an oxygen molar concentration between 100 and 200 μM by fitting $R_1^{\text{PRE:O}_2}$ (Supplementary Note 5). These values are in good agreement with the literature[47].

The use of chelating agents (EDTA, Tris) revealed an additional prominent contribution from another paramagnetic species, which we believe might be transition-metal cation impurities, such as $\text{Mn}^{2+}$ or $\text{Fe}^{2+}$, at μM concentrations. The extra paramagnetic relaxation generates a second shoulder in the NMRD profiles around 0.1 T (samples #H and #I), indicating that the correlation time of the fluctuating intermolecular electron-carbon dipolar interaction is on the order of hundreds of picoseconds. This contribution is quenched by adding Tris or EDTA (or both), as they either chelate the paramagnetic ion $\text{M}^+$, such as $\text{Mn}^{2+}$ or $\text{Fe}^{2+}$, at μM concentrations or shield pyruvate via the CIDER effect[11]. In both cases, the distance to paramagnetic impurities increases, reducing the impact on the relaxation rate (sample #I compared to #F). The low-field limits of these PRE contributions are estimated as $R_1^{\text{PRE:M}^+} = 0.0133 \pm 0.0005\,\text{s}^{-1}$ (samples #I and #E) and $R_1^{\text{PRE:M}^+} = 0.0113 \pm 0.0004\,\text{s}^{-1}$ (samples #H and #D).

Having identified relaxation contributions (Fig. 3A, Supplementary Note 3), we can conclude that for the case of pyruvate in protonated water (sample #I), paramagnetic relaxation contributions become the most important sources of signal loss at low fields, while the high field is dominated by CSA and intermolecular $^1\text{H}$-$^{13}\text{C}$ dipolar couplings.

Using MD simulations, we estimated how increasing temperature reduces both intra- and inter-molecular relaxation contributions, assuming full protonation (Fig. 3B). Likewise, the dipolar contributions due to deuterium coupling can be calculated by scaling the corresponding proton-induced rates with the factor $(I_\text{H}(I_\text{H}+1))/(I_\text{D}(I_\text{D}+1))$ $(\gamma_\text{H}/\gamma_\text{D})^2 \approx 15.9$ as mentioned previously. We estimate that the increase in temperature from 293 K to 353 K can lead to a 2.6-fold decrease in the intermolecular contribution and an averaged 2.8-fold decrease in the intramolecular dipole-dipole contribution. Thus, storing hyperpolarized $^{13}\text{C}$-labeled pyruvate at these temperatures in non-deuterated solvents is predicted to enhance $T_1$ by around 25% at 2 T. When background relaxation sources are eliminated completely, and relaxation occurs exclusively via inter- and intramolecular dipole–dipole interactions, the relaxation time of a fully deuterated system could reach 39 min at 293 K and 110 min at 353 K. In practice, however, contributions from additional mechanisms, such as paramagnetic impurities, partial protonation, or spin–rotation interactions, cannot be completely suppressed, making such long relaxation times experimentally inaccessible. Nevertheless, this estimate highlights the significant gap in our understanding of relaxation contributions and the effective purity of the samples used.

## In vitro validation

To quantify the gain in sensitivity in vitro, we used 5 million HeLa cells in 200 μL PBS mixed with 280 μL of hyperpolarized pyruvate in three dissolution media: (1) optimized DNP sample #C ($\text{D}_2\text{O}$ filtr.), (2) same as sample #C but without radical filtration ($\text{D}_2\text{O}$), and (3) standard DNP sample #G ($\text{H}_2\text{O}$).

The hyperpolarized pyruvate was measured on two systems in parallel: at 1 T without cells and at 9.4 T with HeLa cells. The 1 T measurement was used to directly measure low-field $T_1$ and polarization level. While the filtered sample #C provided slightly longer $T_1$ than the non-filtered sample, as expected ($162.1 \pm 3.9$ s vs $153.8 \pm 12.3$ s), both were about twice as long as the normal sample #G with $77.1 \pm 1.4$ s (Fig. 4A). However, since the acidic pH required for filtration accelerates relaxation, this results in greater signal loss compared to the non-filtered sample #C, while maintaining the same total transfer time (time between dissolution and administration). Therefore, the retained polarization $P(t = 45\,\text{s})$ after a 45 s transfer was ($29.9 \pm 2.4$)%, ($35.6 \pm 2.1$)%, and ($25.8 \pm 2.2$)% for $\text{D}_2\text{O}$ filtr., $\text{D}_2\text{O}$ and $\text{H}_2\text{O}$ samples, respectively (Fig. 4B). The $\text{D}_2\text{O}$ sample retained the most polarization, with $38 \pm 4$% more than the $\text{H}_2\text{O}$ and $19 \pm 2$% more than the $\text{D}_2\text{O}$ filtr. sample.

Upon hyperpolarized pyruvate administration to HeLa cells, we observed production of lactate and evaluated its maximum SNR (see Supplementary Note 1, Supplementary Fig. S2): The achieved values were $227.9 \pm 13.2$, $365.3 \pm 33.5$, and $156.7 \pm 17.9$ for $\text{D}_2\text{O}$ filtr., $\text{D}_2\text{O}$, and $\text{H}_2\text{O}$, respectively (Fig. 4C). Again, the highest SNR of the metabolite was for $\text{D}_2\text{O}$ with a $133 \pm 19$% gain compared to $\text{H}_2\text{O}$ and $60 \pm 7$% compared to $\text{D}_2\text{O}$ filtr.

These results show that, for maximizing $T_1$, the filtration step is necessary, whereas, for maximizing polarization and metabolic SNR, the transfer without radical filtration is more promising. These benefits should also translate to in vivo experiments, where a reasonable SNR is often even more challenging to obtain than in cell experiments. A doubling of SNR would allow reducing the flip angle (if necessary), increasing the measurement duration, or improving measurement accuracy.

We found that the relaxation of [1-$^{13}\text{C}$]pyruvate is affected by sample composition and preparation up to an order of magnitude ($\approx 30$–$300$ s), and up to 100% across magnetic fields (e.g., 50 s to 100 s or 240 s to 70 s). Appropriate sample composition and preparation enabled us to extend the lifetime of [1-$^{13}\text{C}$]pyruvate polarization to 3 min in a typical DNP sample and 4 min in an ideal sample. Previous studies did not observe this effect because experiments were mostly conducted at high fields, whereas relaxation is predominant and relevant for purification and transport at low fields. The effect was deciphered using simulations and can be used to significantly reduce polarization loss during transfer and quality assurance, e.g., at 1 T. These improvements provided real-world gains for metabolic cell experiments, providing an increase in metabolite SNR, $T_1$, and retained polarization. The gain in polarization will further boost the power of hyperpolarized magnetic resonance to probe otherwise inaccessible processes. MD predicted dipole-dipole relaxation rates are lower than experimentally observed, tentatively indicating that even longer relaxation for pyruvate is possible.

## Methods
### Sample compositions
$90.1 \pm 0.5$ mM [1-$^{13}\text{C}$]pyruvate (molecular weight 89.05 g/mol, Sigma-Aldrich, 677175, CAS: 99124-30-8) for non-dDNP samples ($n = 7$) and 98.8 mM for dDNP samples (two samples #C and #G) with pH of $7.6 \pm 0.1$ in 90% $\text{H}_2\text{O}$ (Carl Roth, ID 3478.1) and 10% $\text{D}_2\text{O}$ (Deutero,

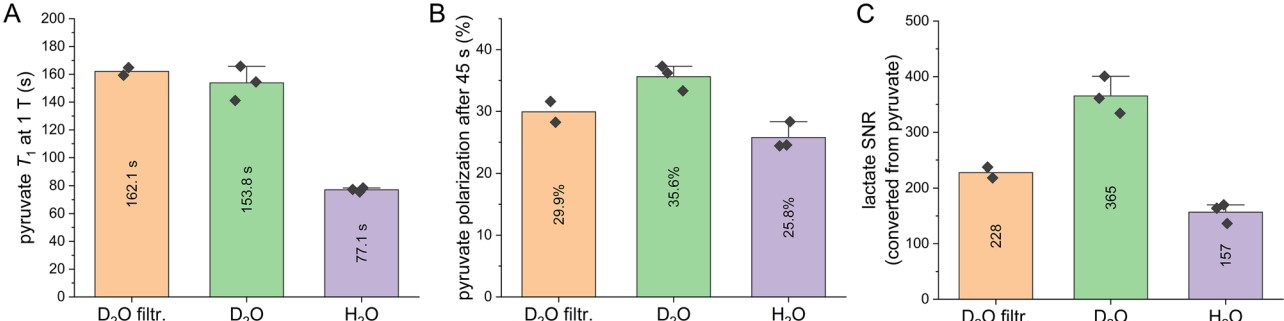

**Fig. 4 | Effect of optimized dissolution media on hyperpolarized [1-¹³C]pyruvate and SNR of its metabolic product lactate in HeLa cell line.** Three dissolution media compositions were tested: D$_2$O filtr. being like sample #C, D$_2$O like #C without the filtration step, and H$_2$O being like sample #G. **A** The retained polarization of pyruvate after a 45 s transfer measured at 1 T without cells was (29.9 ± 1.7)%, (35.6 ± 1.7)%, and (25.8 ± 1.8)% for the samples as introduced. **B** The polarization lifetime of pyruvate, corrected for flip-angle and measured at 1 T without cells, was (162.1 ± 2.8) s, (153.8 ± 10.1) s, and (77.1 ± 1.2) s for the samples as introduced, respectively. **C** The observed maximum SNR for the metabolic product lactate in HeLa cells was (227.9 ± 9.4), (365.3 ± 27.4), and (156.7 ± 14.6), respectively, using the same acquisition parameters. A transfer magnet of about 35 mT was used for D$_2$O filtr. and D$_2$O, but not for H$_2$O. The transfer time was kept constant across all samples to accommodate about 40 s for the transfer, including filtration. D$_2$O and H$_2$O experiments were done on the same day. pH of the dissolution medium (and cells) was 7.95 ± 0.16 (7.65 ± 0.24), 7.38 ± 0.08 (7.10 ± 0.03), and 7.65 ± 0.03 (7.5 ± 0.01), respectively. Two biological repeats were performed for D$_2$O filtr., and three for D$_2$O and H$_2$O, repetition was performed by growing independent flasks for each experiment. H$_2$O is treated as the control group. Respective kinetics in Supplementary Note 1, Supplementary Fig. S2. The bar plots show mean values ± sample SD. Source data are provided as a Source Data file.

00506) or 100% D$_2$O were used. The following chemicals were added: Trityl (Polarize, AH111501), NaCl (Sigma-Aldrich, S98881), Trizma (Tris) buffer (Sigma-Aldrich, T7943), ethylendiamintetracetic acid (EDTA, SERVA, 11280.02). NaOH (Carl Roth, 9356.1) and HCl (TCI Chemicals, H1202) were used to adjust the pH to the desired value.

Filtration of the radical from sample #C was conducted by adding conc. HCl to the dDNP sample after dissolution, followed by filtration of the radical using a 0.2 μm filter (Chromafil Xtra, PET 20/25), and addition of NaOH to re-neutralize the sample. The solution before filtration is yellow-green, while after filtration, it was clear.

Degassing of the samples #A and #B was conducted by three iterations of freezing in liquid N$_2$ followed by vacuuming and slow thawing for non-dDNP samples. The NMR tube was then flame-sealed under vacuum. The dDNP sample #C was degassed for 5 minutes using argon bubbling after dissolution, then closed with a standard NMR cap and parafilm. The D$_2$O dissolution media used for HeLa cells (described below) were degassed for 3 min under argon before dissolution.

A table containing all sample compositions for samples #A-I is supplied in the Supplementary Table S1 in Supplementary Note 1.

### INEPT-enhanced NMRD experiment
The INEPT sequence[38] with refocusing was used without changes and without phase cycling:

$$^{13}C: \quad ---\tau-180_X-\tau-90_X-\tau-180_X-\tau-90_Y-(\text{shuttling})-90_X-\text{FID}$$
$$^{1}H: \quad -90_X-\tau-180_X-\tau-90_Y-\tau-180_X-\tau-----(\text{shuttling})-\text{decoupling}$$

The optimum INEPT interpulse delay, $\tau$, was calibrated for each sample experimentally (Supplementary Fig. S1). Subsequent MFC experiment followed the following sequence: (1) ¹H thermal relaxation at $B_0$ for $D_{B0}$ = 19 s, (2) INEPT sequence to transfer ¹H polarization to longitudinal polarization of [1-¹³C] of pyruvate, (3) shuttling to desired low-field, either inside the NMR bore or in the solenoid electromagnet (SE) within the magnetic shield and with appropriately set current, (4) waiting for ¹³C relaxation at low-field for $D_{LF}$, (5) shuttling to $B_0$ followed by settling delay of 100 ms, (6) ¹³C 90° excitation and signal acquisition with ¹H decoupling.

All extracted NMRD values are shown in the Supplementary Table S2.

The INEPT calibration curve of sample #B is shown in Supplementary Fig. S1.

### Hyperpolarization and NMR signal acquisition
**NMR.** Thermal and hyperpolarized ¹³C MR signals were acquired using a 9.4 T wide-bore NMR (WB400, Avance NEO, Bruker) with a 5 mm BBFO probe. The NMR system temperature was set to 293 K, which is room temperature. Hyperpolarized samples were additionally measured at 1 T ¹³C benchtop NMR (Spinsolve Carbon, Magritek).

**dDNP sample.** Samples for dDNP were prepared by mixing 14 M [1-¹³C] pyruvate with 32 mM trityl radical. 19.5 μL of the mixture was used for samples #C and #G, while 10 μL was used for the cell experiments, to achieve the lower pyruvate concentration required for the cells.

**dDNP dissolution medium.** The dissolution medium was prepared by mixing 1.51 g of Trizma pre-set crystals, 27 mg of EDTA, 0.756 g of NaCl, and 0.81 g of NaOH in 250 mL of water (90% H$_2$O and 10% D$_2$O or 100% D$_2$O). About 3.7 mL of dissolution medium was used for samples #C and #G, and 4 mL for cell experiments. Degassing and addition of 2.5 mM vitamin C (Sigma-Aldrich, A0278) were performed directly before the experiment for samples #C and D$_2$O and D$_2$O filtr. cell experiments. A transfer magnet was used for the optimized DNP dissolution media, like sample #C, but not for standard H$_2$O dissolution, like sample #G. Final concentration in the dissolution medium was 98.9 mM for samples #C and #G, and 42.5 mM for the cell experiments.

### HeLa cell experiments
**Preparation.** The human cervical cancer HeLa cells (from ATCC: CCL-2) were maintained at 37 °C with 5% CO$_2$ in high glucose Dulbecco's Modified Eagle Medium (DMEM; Gibco, Thermo Fisher Scientific, Waltham, MA, USA) containing 4.5 g/L D-glucose, L-glutamine, 1 mM sodium pyruvate (Gibco), 10% fetal bovine serum (FBS South America; Pan-Biotech, Aidenbach, Germany), and 1% penicillin/streptomycin (Gibco). HeLa cells were routinely passaged 2–3 times per week when they reached the target confluency. Two days before each dDNP experiment, three million HeLa cells were seeded in T-75 cell culture

flasks (Sarstedt, Nümbrecht, Germany) and incubated at 37 °C with 5% $CO_2$. On the day of the DNP experiment, the medium was discarded, and cells were washed with Dulbecco's phosphate-buffered saline (PBS; Gibco). Cells were detached with 0.25% trypsin-EDTA solution (Pan-Biotech) and neutralized with DMEM. The cell suspension was centrifuged at 300×g for 5 min, the supernatant was removed, and the cell pellet was resuspended in 2 mL DMEM for cell counting. Five million cells were collected, washed twice with cold PBS, with centrifugation at 300×g for 5 min at 4 °C between washes, and finally resuspended in 200 μL PBS before transfer into a Shigemi NMR tube (Deutero GmbH) for the subsequent DNP experiment.

**Kinetic analysis.** An exponential apodization of 1 Hz was applied for the kinetic and 5 Hz for the polarization analysis. A third-order polynomial baseline correction was applied for each metabolite peak. Phase correction was applied in all cases. Integrals were used to quantify the signals (MNova, 14.2.2, Mestrelab Research S.L.), and SNR was analyzed using the MNova SNR graph feature (Supplementary Fig. S2).

NMR spectra after dissolution were measured in parallel on a 1 T benchtop NMR in a standard NMR tube without cells, and on a 9.4 T NMR in a Shigemi tube with cells to keep the cells within the sensitive area of the coil.

The signal from the first hyperpolarized spectrum at 1 T was used to calculate the liquid-state polarization after sample transfer from dDNP to 1 T NMR. As a reference, a thermally polarized spectrum of the same sample at 1 T was subsequently measured. Liquid-state polarization was not measured at 9.4 T in the same way because the cells continuously consumed the substrate.

## Computational chemistry simulations

Molecular dynamics (MD) simulations were performed using GROMACS[48] to predict longitudinal relaxation rates of pyruvate in water under different temperatures. The initial geometries of both pyruvate and hydrated pyruvate molecules were optimized using Gaussian16[49] with the density functional theory at B3LYP/def2TZVPP level. The optimized structures were then processed through CGenFF[50] to obtain CHARMM36-compatible force field parameters. One molecule of pyruvate and one molecule of hydrated pyruvate were added into a 34*34*34 Å TIP4P water box with the periodic boundary condition applied in all directions. The overall charge of the system was neutralized by $Na^+$ ions. Energy minimization, followed by NVT and NPT equilibration were performed prior to the production run which was run under NPT ensemble. Longitudinal relaxation rates $R_1$ of carboxylate $^{13}C$ of both pyruvate molecules due to $^1H$-$^{13}C$ intra- and intermolecular dipolar couplings were computed with a custom script and averaged over 10 production runs. Results are summarized in Supplementary Table S3. The chemical shift anisotropy was extracted from DFT simulations of the chemical shielding tensor using ORCA[51] at the B3LYP/def2-TZVP level of theory for the pyruvate molecule only.

## Software

TopSpin 4.4.0 (Bruker), SpinsolveExpert 2.02.00 (Magritek), and magnetic-field-cycling hardware and software described in Ref. 35 were used for data collection. Mestrenova 15.1.0-38027 (Mestrelab Research S.L.), OriginPro 9.8.5.201 2021b (OriginLab Corporation) and Mathematica 14.0.0.0 were used for data analysis. GROMACS 2023.3, Gaussian 16RevA.03 and ORCA 5.0.2 software were used for computational chemistry simulations.

## Reporting summary

Further information on research design is available in the Nature Portfolio Reporting Summary linked to this article.

## Data availability

Additional experimental data, fitting, relaxation analysis, cell analysis, and MD simulation results can be found in the supporting information (.pdf).

Source data for Figs. 2–4 are provided with this paper (.xlsx). Raw data generated in this study, together with MD input parameters, DFT calculated geometries, and Mathematica 14 scripts, have been deposited in the Zenodo database[52] under accession code: https://doi.org/10.5281/zenodo.17175125. Source data are provided with this paper.

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

## Author contributions

A.N.P., J.-B.H.: conceptualization, J.P.P.: NMRD experiments, data analysis, E.R.: pyruvate chemical exchange study, F.T., H.Z., A.J.: relaxation simulations, J.P. P., F.T., A.N.P.: preparation of graphics, F.H.M., H.S.: cells cultivation, cells analysis, F.H.M., J. P.P.: dDNP experiments with cells and data analysis, A.N.P., A.J., and J.B.H.: supervision, funding acquisition. All authors contributed to the discussion and interpretation of the results, wrote the original draft, and approved the final version of the paper.

## Funding

We acknowledge funding from the German Federal Ministry of Education and Research (BMBF, 03WIR6208A hyperquant), Heisenberg DFG grant (565789098), DFG (562203308, 555951950, 527469039, HO-4602/2-2, HO-4602/3, HO-4604/6–1, HO-4604/8–1, Inst 257/747-1, EXC2167/2, FOR5042, TRR287). MOIN CC was founded with a grant from the European Regional Development Fund (ERDF) and the Zukunftsprogramm Wirtschaft of Schleswig-Holstein (Project no. 122-09-053). JPP acknowledges funding and support by the Studienstiftung des Deutscher Volkes. AJ acknowledges funding from the US National Science Foundation, award no. CHE 2505792, an award from the ACS PRF 68117-ND6, and an unrestricted gift from Google. FT acknowledges funding from the European Union and Romania through the National Medical Project MySMIS SMIS Code 326475, and the Project ELI-RO/RDI/2024/14 SPARC funded by the Institute of Atomic Physics (Romania). Open Access funding enabled and organized by Projekt DEAL.

## Competing interests

The authors declare no competing interests.
