## [Transparent Peer Review file · Nature Communications]

Over four minutes of pyruvate T1 using chemically and physically induced deceleration of relaxation

Corresponding Author: Dr Andrey Pravdivtsev

Version 0:

Reviewer comments:

Reviewer #1

(Remarks to the Author)

Reviewer comments:

The manuscript addresses an important challenge with respect to the most widely used molecular contrast agent pyruvate: the spin polarization lifetime on ^{13}C of $[1-^{13}\text{C}]$ pyruvate. While there are many individual studies that look at one of the aspects that contribute to relaxation and address this in some way, a study that looks at this in detail and quantifies the individual contributions depending on the magnetic field is timely and a very good idea.

I found myself reading the manuscript with a lot of interest – from an experimentalists point of view. I believe, that this could become a go-to manuscript for many people working on molecular contrast agents from method development and preclinical all the way to hospital applications and maybe even beyond it for people working on other applications where relaxation plays a role. The way this study is done, it could become a gold standard to study relaxation in contrast agents.

That being said, I believe the introduction could be slightly extended to become such a “go-to” manuscript. Therefore, with respect to literature I would suggest additional references in two “directions” (or combining them): hyperpolarization methods and work that addresses relaxation.

Depending on the hyperpolarization method, additional complexities/relaxation sources are introduced. Here, for these additional relaxation sources you focus on dDNP (I assume to its prominence and importance). However, dDNP is mentioned in the manuscript for the first time in the caption of Fig 2, which is quite late. I suggest you introduce it earlier and add a short comment on complexities of alternative methods (such as PHIP SAH).

You use high-field thermal polarization for your relaxation studies, which is a good idea. Nonetheless, many researchers describe how they dealt with relaxation losses when using different hyperpolarization methods, for example buffering solutions or studying degassing, deuteration or magnetic field dependences. While hyperpolarization (and its different variants) is not the focus on your story, many of the ideas here are addressed individually or mentioned in such manuscripts (even though often only as a side note). You are now doing a comprehensive study of the six most important factors - nonetheless consider citing additional work with respect to these factors in your introduction.

In conclusion, I believe that nature communications is a very good fit for this work. The quality of the work is very high. Nonetheless, in the following, I will provide suggestions for minor comments as well as additional questions. I would like to clarify however that the questions come more from a point of genuine interest and not from the lack of detail provided in the current manuscript, which is already rich enough.

Minor comments:

Line 31: maybe rephrase “must account for time for”

Line 71 “and the methyl group” You mean the deuteration of the methyl group I assume? Maybe rephrase to avoid ambiguity

Line 78. If your first sentence of your discussion is about the automation, a short description of the automation is a good idea. (even though it is published). Alternatively start your section with your story (that you systematically studied different parameters and acquired over 4300 FIDs.

Line 90: “(Fig. 2#H)” you are comparing two plots (#H and #I here). Is suggest to refer to both when doing so.

Line 106: Can this conclusion be drawn in this way? Would you not need a measurement at the same conditions but only with EDTA and without Tris?

Line 107: You are comparing #D and #I to discuss the chelating effect. Due to the effect of D₂O vs H₂O, comparing to #D to #H, would make more sense.

Line 114: Fig. 2. This is a beautiful figure and summary of your main experimental findings. If I understood it correctly from the main manuscript, you assume an exponential decay for the relaxation. In the SI you specify that you are fitting biexponentially. Have you seen types of decays? Can you comment on this finding in the main text?

Line 134: the wording "negligible" is slightly exaggerated

Line 140: Can you comment on how biocompatible solutions with Tris, EDTA and D₂O are?

Line 145: Clarify that the Trityl Radicals are part of the DNP process to make it easier for readers from other fields to understand why they are here. (and why radical filtration is necessary). Do the same for Vitamin C.

Also, when comparing #G and #I, a key difference is adding EDTA and Tris. This section could therefore fit well right after the chelation discussion (around line 112).

Line 153: Your 2.6- fold increase in SNR considering 1 min transfer time is impressive. At UCSF (where the HTMRC is also hosted) they record typical transfer times over hundreds of human experiments. Can you comment and compare to their timing the manuscript?

Line 161: "limiting the T₁" either just "limiting T₁" or "limiting the T₁ relaxation time"

Line 162: "in ¹³C spectrum" "in the ¹³C spectrum". Also, while its clear what is meant after reading this sentence (and while it is important), the sentence is quite long and a bit unintuitive to read, maybe consider moving the chemical shift information into a bracket at the end of the sentence or into an individual sentence together with the ratio at neutral pH?

Line 177: Fig3: Additional experiments might be quite some work and outside of the scope of this manuscript, but have you measured or could you now, that you looked at all the relaxation contributions individually, estimate other "combinations"/conditions than the ones you measured? For example, when moving this towards applications, the overall costs may play a role at some point. And the deuteration adds a significant part of that (both deuteration of pyruvate as well as the solvent). Thus, a scenario where only degassing but not deuteration is considered may be interesting for some applications. Have you tried that? Can you estimate T₁'s of other scenarios based on your MD-simulations?

Line 189: Can you comment/reference to how to get to this estimation?

Line 219: "The maximum value of T₁ is approximately 2 T" "The maximum value of T₁ is approximately at 2 T"

Line 242: To see the temperature dependence experimentally would of course also be interesting but beyond the scope of this work.

Line 246: Biocompatibility will also be a challenge at the temperatures.

SI:

Chapter 1: Why the different concentration for the dDNP samples?, Why different degassing methods? define n=2? Does the salt in the sample that you introduce when neutralizing impact your relaxation?

Chapter 4, Minor language change suggestions:

line 1. the pyruvate solution;

line 8: The PyH form...

Reviewer #2

(Remarks to the Author)

Page 2, line 54. 'CIDER'. Please briefly explain what CIDER is for readers who are not familiar with reference 11.

Page 5, line 149. '60s'. This seems excessive compared to reported values, which are usually around 30s. Could the author give a reference?

Page 6, line 207. 'we can'. No need to switch from the past to the present tense here.

Page 7, line 251. 'relaxation then'. Add a comma.

Page 7, line 252. 'relaxation times the order'. There is a problem with the sentence.

Reviewer #3

(Remarks to the Author)

¹⁻¹³C pyruvate is one of the most extensively studied metabolites in the hyperpolarization field and has shown great promise as an agent for both cancer diagnosis and therapeutic monitoring. Its longer relaxation time is particularly advantageous for probing metabolic pathways related to glycolysis. Numerous studies have attempted to extend the relaxation time of ¹⁻¹³C pyruvate through strategies such as pyruvate deuteration, use of deuterated solvents, lowering magnetic field strength, sample degassing, and other approaches, each achieving varying degrees of success.

This study is distinctive because the authors systematically evaluated six specific factors to enhance the relaxation constant:

- (1) chelating agent (EDTA),
- (2) buffer (Tris),
- (3) solvent deuteration (D₂O),
- (4) sample degassing (oxygen removal),
- (5) pyruvate deuteration, and
- (6) applied magnetic field strength.

The research is well-conducted, and the improvements in relaxation time represent a meaningful advancement in the field. However, the manuscript requires additional data to fully demonstrate the practical significance of these findings. In particular, the following major points should be addressed:

1. The increased relaxation time of ¹⁻¹³C pyruvate under the optimized conditions should be demonstrated in an in-vitro cell line model.
2. Additionally, the authors should show that the extended relaxation time of ¹⁻¹³C pyruvate improves performance in an in-vivo model, specifically addressing current limitations arising from the comparatively short relaxation times.

Version 1:

Reviewer comments:

Reviewer #1

(Remarks to the Author)

Dear authors,

an excellent manuscript that can be published as is now. On top of all the work bringing this to a go-to manuscript on pyruvate relaxation and lifetime you even added in vitro data.

PS: One comment on the in vitro SI section: Well explained. There is no need for a space before °C though.

Reviewer #2

(Remarks to the Author)

Reviewer #3

(Remarks to the Author)

The authors have adequately addressed the reviewers' concerns, and the manuscript can be accepted for publication.

REVIEWER**COMMENTS****Reviewer #1 (Remarks to the Author):**

The manuscript addresses an important challenge with respect to the most widely used molecular contrast agent pyruvate: the spin polarization lifetime on ^{13}C of $[1-^{13}\text{C}]$ pyruvate. While there are many individual studies that look at one of the aspects that contribute to relaxation and address this in some way, a study that looks at this in detail and quantifies the individual contributions depending on the magnetic field is timely and a very good idea. I found myself reading the manuscript with a lot of interest – from an experimentalists point of view. I believe, that this could become a go-to manuscript for many people working on molecular contrast agents from method development and preclinical all the way to hospital applications and maybe even beyond it for people working on other applications where relaxation plays a role. The way this study is done, it could become a gold standard to study relaxation in contrast agents.

Answer: Thank you very much for acknowledging the importance of understanding the relaxation mechanisms of the hyperpolarization probes used today and in future ones. We believe that this work on pyruvate T_1 prolongation showcases the power of unravelling the mechanisms of nuclear relaxation with precise implications for basic research, pre-clinical, and clinical studies alike. We agree that it is only the beginning of such quantitative studies of molecular contrast agents.

That being said, I believe the introduction could be slightly extended to become such a “go-to” manuscript. Therefore, with respect to literature I would suggest additional references in two “directions” (or combining them): hyperpolarization methods and work that addresses relaxation.

Answer: Thank you, we have added references on common hyperpolarization methods and on work addressing relaxation. In no means is the hyperpolarization and relaxation review complete, but we hope that it gives sufficient minimum for the reader.

Changes:

The currently most prominent hyperpolarization methodology is dissolution dynamic nuclear polarization (dDNP)¹⁴, with many clinical examples to date^{15,16}. Parahydrogen-induced polarization (PHIP)^{17,18} and Signal amplification by reversible exchange (SABRE)^{19,20} are rapidly developing viable alternatives for liquid-state hyperpolarization. Pyruvate is arguably one of the most frequently used hyperpolarization agents^{21,10}. Intriguingly, the relaxation lifetime of pyruvate was reported to differ across hyperpolarization methods²²; however, the sample composition varied slightly. Since the polarization method should not affect the relaxation time, we investigated how the sample composition affects the observed relaxation time of pyruvate. We identified the optimal composition and tested it *in vitro* following dDNP hyperpolarization. The detailed challenges, comparisons, and state of the art of the hyperpolarization method have been reviewed previously^{23–26}. Each hyperpolarization method has its challenges, such as filtration of paramagnetic impurities in dDNP or of metal-organic catalysts or solvents in PHIP/SABRE. The ultimately prepared and purified sample should have the same hyperpolarization lifetime.

We systematically investigated the sample composition of $[1-^{13}\text{C}]$ pyruvate and identified conditions under which the relaxation time exceeds 4 minutes. We identified at least six factors to achieve this: chelation agent (EDTA), buffer (Tris), solvent deuteration (D_2O), sample degassing (no O_2), pyruvate deuteration, and the applied magnetic field (**Fig. 1**), many of which have been explored individually in the scope of relaxation studies and hyperpolarization experiments before^{8,11,27–32}. For

example, previously measured high-field T_1 of $[1-^{13}\text{C}]$ pyruvate did not reveal a significant impact of deuteration on its T_1 ²⁹, despite the fact that deuteration positively impacts the relaxation properties of remaining spins^{30,31}. As shown below, the achieved relaxation value results from a compounding effect of the systematic deactivation of relaxation sources. The effect of chemically induced deceleration of nuclear relaxation (CIDER) was recently discovered, demonstrating that hydrogen-bonding additives can reduce relaxation induced by rapid chemical exchange of quickly relaxing species or by increasing the distance to paramagnetic impurities; it likely plays a key role in explaining the impact of Tris buffer¹¹. Tris and EDTA are commonly used in clinical dDNP experiments^{21,33} due to their biocompatibility. In addition, D₂O has recently been shown to be safe for human imaging^{32,34}. The use of D₂O prolongs T_1 relaxation times and preserves polarization levels, thereby increasing the signal-to-noise ratio (SNR) of pyruvate and its metabolite lactate in humans³².

Depending on the hyperpolarization method, additional complexities/relaxation sources are introduced. Here, for these additional relaxation sources you focus on dDNP (I assume to its prominence and importance). However, dDNP is mentioned in the manuscript for the first time in the caption of Fig 2, which is quite late. I suggest you introduce it earlier and add a short comment on complexities of alternative methods (such as PHIP SAH).

Answer: Indeed, this is a valid point and helps to clarify the scope of the results presented in this work. At the same time, the results obtained using thermally polarized samples should have broader applications beyond dDNP. We hope that the current introduction reflects our ideas as well as introduces the objectives and validation approach.

Added text in the introduction:

The currently most prominent hyperpolarization methodology is dissolution dynamic nuclear polarization (dDNP)¹⁴, with many clinical examples to date^{15,16}. Parahydrogen-induced polarization (PHIP)^{17,18} and Signal amplification by reversible exchange (SABRE)^{19,20} are rapidly developing viable alternatives for liquid-state hyperpolarization. Pyruvate is arguably one of the most frequently used hyperpolarization agents^{21,10}. Intriguingly, the relaxation lifetime of pyruvate was reported to differ across hyperpolarization methods²²; however, the sample composition varied slightly. Since the polarization method should not affect the relaxation time, we investigated how the sample composition affects the observed relaxation time of pyruvate. We identified the optimal composition and tested it *in vitro* following dDNP hyperpolarization. The detailed challenges, comparisons, and state of the art of the hyperpolarization method have been reviewed previously^{23–26}. Each hyperpolarization method has its challenges, such as filtration of paramagnetic impurities in dDNP or of metal-organic catalysts or solvents in PHIP/SABRE. The ultimately prepared and purified sample should have the same hyperpolarization lifetime.

You use high-field thermal polarization for your relaxation studies, which is a good idea. Nonetheless, many researchers describe how they dealt with relaxation losses when using different hyperpolarization methods, for example buffering solutions or studying degassing, deuteration or magnetic field dependences. While hyperpolarization (and its different variants) is not the focus on your story, many of the ideas here are addressed individually or mentioned in such manuscripts (even though often only as a side note). You are now doing a comprehensive study of the six most important factors -nonetheless consider citing additional work with respect to these factors in your introduction.

Answer: This is absolutely correct; it was our aim to present a comprehensive study of all the individual components. We tried to collect examples from all variants.

Modified text:

We systematically investigated the sample composition of [1-¹³C]pyruvate and identified conditions under which the relaxation time exceeds 4 minutes. We identified at least six factors to achieve this: chelation agent (EDTA), buffer (Tris), solvent deuteration (D₂O), sample degassing (no O₂), pyruvate deuteration, and the applied magnetic field (Fig. 1), many of which have been explored individually in the scope of relaxation studies and hyperpolarization experiments before^{8,11,27–32}. For example, previously measured high-field T_1 of [1-¹³C]pyruvate did not reveal a significant impact of deuteration on its T_1 ²⁹, despite the fact that deuteration positively impacts the relaxation properties of remaining spins^{30,31}.

In conclusion, I believe that nature communications is a very good fit for this work. The quality of the work is very high. Nonetheless, in the following, I will provide suggestions for minor comments as well as additional questions. I would like to clarify however that the questions come more from a point of genuine interest and not from the lack of detail provided in the current manuscript, which is already rich enough.

Answer. Thank you for this assessment. We are happy about you being interested in this work, which we find very exciting as well. We answered all minor questions in as much detail as possible, without overloading the manuscript.

Minor

comments:

Line 31: maybe rephrase “must account for time for”

Answer: Valid point, the text was rephrased.

Modified text:

The first period is primarily determined by physiology, whereas the second period must allow time for quality control, sample transfer, administration, *in vivo* distribution, and tracing the desired function, without the polarization decaying below the detection threshold.

Line 71 “and the methyl group” à You mean the deuteration of the methyl group, I assume? Maybe rephrase to avoid ambiguity

Answer: Nicely spotted, the text was rephrased.

Modified text:

Degassing the sample, solvent deuteration, and deuteration of the methyl group further prolong the relaxation time.

Line 78. If your first sentence of your discussion is about the automation, a short description of the automation is a good idea. (even though it is published). Alternatively start your section with your story (that you systematically studied different parameters and acquired over 4300 FIDs).

Answer: We have slightly altered the story to better motivate why we used the newly developed MFC system for this study and added a pre-face for results and discussion to guide the reader

Modified text:

Results and Discussion

The Results and Discussion are organized into three parts. First, we introduce the methodology used to measure pyruvate relaxation as a function of magnetic field and sample composition. Second, we quantitatively dissect the individual relaxation contributions and demonstrate how controlled variations in sample composition enabled their identification and mitigation. This analysis is supported by molecular dynamics and DFT simulations of relevant relaxation parameters. Finally, we validate the

optimized sample composition *in vitro* by demonstrating improved metabolic sensitivity resulting from prolonged relaxation times.

Magnetic field cycling experiments

To understand the impact of sample composition and magnetic field on the relaxation properties of [1-¹³C]pyruvate, we measured relaxation at thermal equilibrium (i.e., without hyperpolarization) using an MFC system. We varied sample composition, as discussed below, and acquired more than 4,300 free induction decay (FID) transients to systematically probe the nuclear relaxation across a wide range of well-controlled parameters (Fig. 2). The previously developed, highly automated MFC system³⁶ was therefore essential, performing approximately 8,600 back-and-forth sample shuttling operations during data acquisition. This experimental series yielded NMRDs at nine different conditions, as explained in Fig. 2. T_1 measurements at $B_0 = 9.4$ T were done without shuttling.

Line 90: "(Fig. 2#H)" à you are comparing two plots (#H and #I here). Is suggest to refer to both when doing so.

Answer: Good point, we have added references to both figures for clarification.

Changes:

Substitution of H₂O (Fig. 2#I) with D₂O (#H) increased T_1 uniformly across the entire field range by ≈ 15 s.

Line 106: Can this conclusion be drawn in this way? Would you not need a measurement at the same conditions but only with EDTA and without Tris?

Answer: A valid question. We drew this conclusion based on the fact that the addition of EDTA did not change the low-field T_1 by any meaningful amount when Tris was already present; however, we do know that the addition of Tris alone increased T_1 from 30.9 to 50.0 in H₂O. It therefore implies that Tris protects pyruvate (possibly through a CIDER-like interaction) from the same mechanisms that EDTA targets through its well-known chelating capabilities, or, conversely, that EDTA doesn't reduce any additional relaxation channels compared to Tris (in H₂O). If EDTA alone were as efficient as Tris, we could not definitely say from this data alone. Considering the number of possible sample compositions, we selected the bare minimum to guide the reader to this sample. We are currently studying these effects in greater detail across different variants, including varying concentrations, but the results are still being collected and analysed.

Modified text:

To further investigate a possible chelating effect, we compared Tris-buffered solutions with and without EDTA (Fig. 2#E compared to #F). Adding EDTA increased T_1 only marginally (1–3 s), suggesting that Tris alone protects against a substantial fraction of the relaxation channels that EDTA targets; EDTA is a well-known chelation agent that should protect against metal cations.

Line 107: You are comparing #D and #I to discuss the chelating effect. Due to the effect of D₂O vs H₂O, comparing to #D to #H, would make more sense.

Answer: Indeed, adding the comparison to the deuterated sample makes sense. We initially wanted to showcase the total gain relative to the "bare bones" sample #I, but we've now added a comparison with #H as well.

Modified text:

To further investigate a possible chelating effect, we compared Tris-buffered solutions with and without EDTA (Fig. 2#E compared to #F). Adding EDTA increased T_1 only marginally (1–3 s), suggesting that Tris alone protects against a substantial fraction of the relaxation channels that EDTA targets; EDTA is a well-known chelation agent that should protect against metal cations. Combining

Tris and EDTA in D₂O in #D resulted in a T_1 increase of ≈ 50 s at low fields and ≈ 20 s at high fields compared to #I, or ≈ 40 s at low fields and ≈ 10 s at high fields compared to #H, which is also D₂O sample as #D but lacks Tris and EDTA, exceeding ≈ 115 s in the intermediate regime.

Line 114: Fig. 2. This is a beautiful figure and summary of your main experimental findings. If I understood it correctly from the main manuscript, you assume an exponential decay for the relaxation. In the SI you specify that you are fitting biexponentially. Have you seen types of decays? Can you comment on this finding in the main text?

Answer: We are happy that you like the figure 2. Nicely spotted, indeed we found a smaller and much faster relaxing component $T_{1,2}$ which we found to improve the fit of the data. We did not discuss it in the main text, as the conclusions drawn from the second component are not clear. Typically, our sample size is bigger than the area of the NMR coil, so following INEPT enhancement we may expect some diffusion of non-enhanced spins to diffuse into the sensitive area of the coil, observable as another relaxation component.

Added text to SI:

Interestingly, we found a second “relaxation” component during NMRD measurements of pyruvate (see examples in the SI, Fig. S3). This second relaxation component was always much faster than the reported ¹³C relaxation time here: it ranged from 1.1 to 8.2 (Tab. S2). The longest second relaxation component was found for sample #B, which also has a long T_1 . For sample #A no two-exponential behavior was observed, hence it was fit with a regular one-exponential curve. We had two hypotheses on the origin of the effect: diffusion of the sample during field cycling and polarization transfer from protons to ¹³C. However, the measurements do not confirm this effect, as the amplitude was shown to be deuteration-independent (hence no polarization transfer from protons), and it did not show a perfect correlation with the relaxation time (hence unlikely diffusion). This component was typically less than 2% of the main component's amplitude and could therefore be ignored by skipping some of the first points of kinetics. However, for simplicity, we opted for the two-exponential fitting. When checking for the difference between two- and one-exponential fitting, an average absolute deviation in T_1 of only $(2.36 \pm 2.14)\%$ was observed across all samples and fields.

Fig. S3: Slight two-exponential behavior of samples #H and #D from Fig. 2 in the main text. A slight two-exponential behavior can be observed for the first few points. The amplitude of the second component is typically less than 2% of the first exponential component.

Line 134: the wording “negligible” is slightly exaggerated

Answer: We have exchanged “negligible” with “small”.

Line 140: Can you comment on how biocompatible solutions with Tris, EDTA and D2O are?

Answer: Tris and EDTA are the most commonly used additives for hyperpolarized pyruvate by dDNP, as outlined by reviews on clinical application and the components of the dissolution medium mentioned in clinical studies themselves. For D2O, there was a toxicity study for hyperpolarized pyruvate, which showed no negative effects. The text was modified to include this information.

Added text in the introduction:

Tris and EDTA are commonly used in clinical dDNP experiments^{21,33} due to their biocompatibility. In addition, D₂O has recently been shown to be safe for human imaging^{32,34}. The use of D₂O prolongs T₁ relaxation times and preserves polarization levels, thereby increasing the signal-to-noise ratio (SNR) of pyruvate and its metabolite lactate in humans³².

Line 145: Clarify that the Trityl Radicals are part of the DNP process to make it easier for readers from other fields to understand why they are here. (and why radical filtration is necessary). Do the same for Vitamin C.

Answer: You are right, it is probably a good idea to inform the reader about these aspects. We have added a section in the text to explain these details.

Added text in discussion:

Radicals, such as Trityl AH11501, are required for the dDNP process as a source of polarization. While chelating agents such as EDTA are typically used to remove ionic impurities from the solution, vitamin C may be used as a redox radical scavenger to lessen the impact of the radicals used for the DNP process⁴³ (in liquid state, see below).

Also, when comparing #G and #I, a key difference is adding EDTA and Tris. This section could therefore fit well right after the chelation discussion (around line 112).

Answer: The mentioned section was moved up as suggested.

Line 153: Your 2.6- fold increase in SNR considering 1 min transfer time is impressive. At UCSF (where the HTMRC is also hosted) they record typical transfer times over hundreds of human experiments. Can you comment and compare to their timing the manuscript?

Answer: Our reported transfer time of ~60 s is consistent with clinically implemented hyperpolarized ¹³C workflows at UCSF, where dissolution-to-injection times on the order of ~1 min have been reported (e.g., 63 ± 4 s in Chen et al., 2020). Differences between sites reflect local physical layouts (polarizer-to-scanner distance), QC/release steps, and injection coordination; we did not find a publication that cites typical transfer times across hundreds of experiments, unfortunately.

Reference was added and the text was modified.

Modified text:

Starting with 100% polarization, and assuming a period of 60 s between polarization and observation that should be sufficient for quality assurance and administration in a clinical setting^{32,44}, ...

Line 161: “limiting the T1” à either just “limiting T1” or “limiting the T1 relaxation time”

Answer: “The” was removed.

Line 162: “in 13C spectrum” à “in the 13C spectrum”. Also, while its clear what is meant after

reading this sentence (and while it is important), the sentence is quite long and a bit unintuitive to read, maybe consider moving the chemical shift information into a bracket at the end of the sentence or into an individual sentence together with the ratio at neutral pH?

Answer: “The” was added. The sentence was split into two sentences.

Modified text:

First, we highlight two primary peaks in the ^{13}C spectrum corresponding to the oxo (Py) and hydrated (PyH) pyruvate forms with a ~ 8.4 ppm chemical shift difference at neutral pH and at room temperature. Under these conditions, their respective chemical shifts are ~ 170.3 ppm (Py) and ~ 178.7 ppm (PyH) with a ratio of $p_{\text{Py}}:p_{\text{PyH}} \sim 10:1$ (see SI)⁴⁰.

Line 177: Fig3: Additional experiments might be quite some work and outside of the scope of this manuscript, but have you measured or could you now, that you looked at all the relaxation contributions individually, estimate other “combinations”/conditions than the ones you measured? For example, when moving this towards applications, the overall costs may play a role at some point. And the deuteration adds a significant part of that (both deuteration of pyruvate as well as the solvent). Thus, a scenario where only degassing but not deuteration is considered may be interesting for some applications. Have you tried that?

Can you estimate T_1 's of other scenarios based on your MD-simulations?

Answer: We tested only a fraction of parameters: NaOH, Tris, EDTA, D_2O , O_2 , d_4 . For the fixed concentrations the total number of combinations is $6!$, and more if one would like to understand the impact of concentration. We are currently collecting more data, but adding one more combination/experiment would not be complete, nor would it add clarity to the manuscript.

Since the contributions are additive, one does not need MD; one simply needs experimental results. By comparing #B and #D, you can see the improvement of R_1 is from $\sim 1/90$ s until $1/180$ s at low fields, meaning the delta R_1 is about $1/180$ s, which is the oxygen contribution. If one now takes the sample E, with R_1 about $1/50$ s, then degassing could elevate the R_1 to $1/50 - 1/180 \sim 2/180 \sim 1/70$ s. So, this gives good estimates. As you see, the benefits are not as exciting as in the case of D_2O . Considering this we added this discussion to the text.

Modified text:

Considering the additive nature of the relaxation sources, one can estimate the positive impact from degassing H_2O sample #E: oxygen relaxation contribution at low fields is about $1/84$ s – $1/188$ s ~ 0.006 s $^{-1}$, which would result in an increase of relaxation time for sample #E from 51.5 s to $1/(1/51.5 \text{ s} - 0.006 \text{ s}^{-1}) \sim 74$ s. This 20 s gain is not as impressive as the over 100 s gain in D_2O , indicating that without solvent or substrate deuteration, the over 100 s T_1 remains challenging.

Line 189: Can you comment/reference to how to get to this estimation?

Answer: We hope the previous answer also addresses this question, as we are not sure of the content. The discussion on the contributions of different sources assumes that, once you know all contributions, you can add them as relaxation sources, which are considered independent (to a high order of accuracy).

Line 219: “The maximum value of T_1 is approximately 2 T” à “The maximum value of T_1 is approximately at 2 T”

Answer: Thanks, we have added “approximately reached at 2 T”.

Line 242: To see the temperature dependance experimentally would of course also be interesting but beyond the scope of this work.

Answer: We totally agree. The current limitation of the MFC cycling setup is the temperature control: while at B_0 the Bruker spectrometer's temperature control unit precisely maintains the sample temperature, we did not achieve temperature control at lower fields. This is in the plans, but it is far from realisation; therefore, only room-temperature studies are currently possible.

Line 246: Biocompatibility will also be a challenge at the temperatures.

Answer: Thank you for your comment. Indeed, the most compatible administration would be at body temperature, which already provide a relaxation benefit over the 17°C colder room temperature that we measured in this work. Higher temperatures must lead to longer T_1 , in fact, which we measured at high fields but not at low fields for the reasons described above. Administration at room to body temperature or slightly above is perfectly acceptable, higher temperatures may not be biocompatible, but still can be advisable for storage, if necessary.

SI:

Chapter 1: Why the different concentration for the dDNP samples?

Answer: A valid question. For the NMRDs, we “chose” about 90 mM as the desired concentration as a middle ground between lower concentrations used, e.g., for cell experiments, and higher concentrations, e.g., used for in vivo experiments. The DNP sample's concentration was determined by the amount of NaOH used for neutralization in the dissolution medium, which was optimized to 100 mM. We agree that it would have been more reasonable to adjust the NaOH concentration; we wanted to keep our standardized processes intact, assuming that a 10% change in pyruvate concentration would not significantly alter the relaxation behaviour.

Why different degassing methods?

Answer: In general, we wanted to test the impact of oxygen removal without introducing additional molecules (such as argon). However, the dissolution medium is inevitably in contact with air and cannot be kept under vacuum (unlike the flame-sealed NMR tubes). Also, the dissolution module is pressurized with helium, and following the dissolution, helium is used to expel the remaining liquid. We therefore reverted to argon-based oxygen removal for the DNP experiments, which should go hand in hand with the noble gas used throughout the DNP dissolution and the atmospheric pressure environment.

Modified text in the main text:

An important methodological difference concerns the degassing procedures. Sample #C was degassed by 3 minutes of argon bubbling, whereas samples #A and #B were degassed using three freeze–pump–thaw cycles. We used argon bubbling for #C (and the other dDNP samples discussed below) because the dissolution medium cannot be maintained under vacuum during loading in DNP, unlike the flame-sealed NMR tubes used for #A and #B. Freeze–pump–thaw degassing would leave the solution undersaturated with gas, leading to rapid oxygen uptake upon exposure to air. Argon bubbling removes dissolved oxygen while maintaining gas saturation, making oxygen removal more sustainable and suitable for dissolution media in dDNP, aqueous solution in PHIP, and SABRE experiments. In contrast, freeze–pump–thaw cycles are more appropriate for sealed thermal relaxation studies, as it does not introduce additional substances.

define n=2?

Answer: “n=2” was exchanged with “two samples #C and #G”.

Does the salt in the sample that you introduce when neutralizing impact your relaxation?

Answer: We cannot say this definitely; we included the salt to match the dissolution medium composition to the clinical setting, where NaCl may be added for isotonicity. This will certainly be important to investigate in the future.

Chapter 4, Minor language change suggestions:

line 1. the pyruvate solution;

Answer: "The" was added to the sentence.

line 8: The PyH form...

Answer: "The" was added to the sentence.

Reviewer #2 (Remarks to the Author):

Page 2, line 54. 'CIDER'. Please briefly explain what CIDER is for readers who are not familiar with reference 11.

Answer: Thank you for your comments and thoughts on the manuscript. We have modified this section in the manuscript to introduce CIDER with some explanation.

Modified text:

We systematically investigated the sample composition of [1-¹³C]pyruvate and identified conditions under which the relaxation time exceeds 4 minutes. We identified at least six factors to achieve this: chelation agent (EDTA), buffer (Tris), solvent deuteration (D₂O), sample degassing (no O₂), pyruvate deuteration, and the applied magnetic field (**Fig. 1**), many of which have been explored individually in the scope of relaxation studies and hyperpolarization experiments before^{8,11,27–32}. For example, previously measured high-field T_1 of [1-¹³C]pyruvate did not reveal a significant impact of deuteration on its T_1 ²⁹, despite the fact that deuteration positively impacts the relaxation properties of remaining spins^{30,31}. As shown below, the achieved relaxation value results from a compounding effect of the systematic deactivation of relaxation sources. The effect of chemically induced deceleration of nuclear relaxation (CIDER) was recently discovered, demonstrating that hydrogen-bonding additives can reduce relaxation induced by rapid chemical exchange of quickly relaxing species or by increasing the distance to paramagnetic impurities; it likely plays a key role in explaining the impact of Tris buffer¹¹. Tris and EDTA are commonly used in clinical dDNP experiments^{21,33} due to their biocompatibility. In addition, D₂O has recently been shown to be safe for human imaging^{32,34}. The use of D₂O prolongs T_1 relaxation times and preserves polarization levels, thereby increasing the signal-to-noise ratio (SNR) of pyruvate and its metabolite lactate in humans³².

Page 5, line 149. '60s'. This seems excessive compared to reported values, which are usually around 30s. Could the author give a reference?

Answer: Indeed, it is important to accurately consider the time between dissolution and injection. We found only limited reports in the literature on exact injection times in a clinical setting, with values typically ranging from about 60 s to 90 s. We have added references that match experiences when talking to sites in clinical settings. A discussion of shorter transfer durations was also added.

Modified text (and added references):

Starting with 100% polarization, and assuming a period of 60 s between polarization and observation that should be sufficient for quality assurance and administration in a clinical setting^{32,44}, and a magnetic field profile between dDNP system and NMR measured for our settings before (Ref. ³⁶, **Fig. 6**, magnetic profile without transfer magnet), 27.4% will be retained for suboptimal (**#G**) and 70.4% for optimal (**#C**) dDNP samples. Hence, optimizing the sample composition could lead to a 2.6-times increase in SNR. For shorter transfer durations during non-human experiments (such as 19.5 s in Ref. ³⁶) the benefit will be lower, with 66% retained for suboptimal (**#G**) and 89% for optimal (**#C**) dDNP samples, therefore SNR is about 1.4-times higher for **#C**.

Page 6, line 207. 'we can'. No need to switch from the past to the present tense here.

Answer: Indeed, thank you for pointing this out.

Modified text:

... we were able to determine ...

Page 7, line 251. 'relaxation then'. Add a comma.

Answer: Thanks, a comma was added.

Page 7, line 252. 'relaxation times the order'. There is a problem with the sentence.

Answer: We think there is some confusion here, if you are referring to this sentence:

"Probably at such a long relaxation times the other relaxation mechanisms would be dominated by paramagnetic impurities"

Where "other" was possibly confused with "order". Still, we re-arranged the sentence to make it clearer:

Modified text:

When background relaxation sources are eliminated completely, and relaxation occurs exclusively via inter- and intramolecular dipole–dipole interactions, the relaxation time of a fully deuterated system could reach 39 min at 293 K and 110 min at 353 K. In practice, however, contributions from additional mechanisms, such as paramagnetic impurities, partial protonation, or spin–rotation interactions, cannot be completely suppressed, making such long relaxation times experimentally inaccessible. Nevertheless, this estimate highlights the significant gap in our understanding of relaxation contributions and the effective purity of the samples used.

Reviewer #3 (Remarks to the Author):

$1\text{-}^{13}\text{C}$ pyruvate is one of the most extensively studied metabolites in the hyperpolarization field and has shown great promise as an agent for both cancer diagnosis and therapeutic monitoring. Its longer relaxation time is particularly advantageous for probing metabolic pathways related to glycolysis. Numerous studies have attempted to extend the relaxation time of $1\text{-}^{13}\text{C}$ pyruvate through strategies such as pyruvate deuteration, use of deuterated solvents, lowering magnetic field strength, sample degassing, and other approaches, each achieving varying degrees of success.

This study is distinctive because the authors systematically evaluated six specific factors to enhance the relaxation constant:

- (1) chelating agent (EDTA),
- (2) buffer (Tris),
- (3) solvent deuteration (D_2O),
- (4) sample degassing (oxygen removal),
- (5) pyruvate deuteration, and
- (6) applied magnetic field strength.

The research is well-conducted, and the improvements in relaxation time represent a meaningful advancement in the field.

Answer: Thank you for your thoughts on the work, time, and acknowledging the advancements for the field, which we think are inevitable if we want to drive hyperpolarization towards clinical application.

However, the manuscript requires additional data to fully demonstrate the practical significance of these findings. In particular, the following major points should be addressed:

1. The increased relaxation time of $1\text{-}^{13}\text{C}$ pyruvate under the optimized conditions should be demonstrated in an in-vitro cell line model.

Answer: Thank you for pointing this out. We agree that this is a relevant experiment to showcase the benefits of optimizing T_1 and could be a nice demonstration. We have conducted additional experiments showcasing the benefits of optimizing the dissolution media in real-world experiments. We demonstrated an increase of 38% more retained polarization compared to a standard sample and more than a doubling in metabolite (lactate) SNR following injection of hyperpolarized pyruvate to HeLa cells. We think that this demonstration is sufficient, but definitely more studies (not just a demonstration) would also be beneficial.

Added text in the main manuscript:

***In vitro* validation**

To quantify the gain in sensitivity *in vitro*, we used 5 million HeLa cells in 200 μL PBS mixed with 280 μL of hyperpolarized pyruvate in three dissolution media: 1) optimized DNP sample #C (D_2O filtr.), 2) same as #C but without radical filtration (D_2O), and 3) standard DNP sample #G (H_2O).

The hyperpolarized pyruvate was measured on two systems in parallel: at 1 T without cells and at 9.4 T with HeLa cells. The 1 T measurement was used to directly measure low-field T_1 and polarization level. While the filtered sample #C provided slightly longer T_1 than the non-filtered sample, as expected (162.1 ± 2.8 s vs 153.8 ± 10.1 s), both were about twice as long as the normal sample #G with 77.1 ± 1.2 s (Fig. 4A). However, since the acidic pH required for filtration accelerates relaxation, this results in greater signal loss compared to the non-filtered sample #C, while maintaining the same total transfer time (time between dissolution and administration). Therefore, the retained polarization $P(t=45$ s) after a 45 s transfer was $(29.9 \pm 1.7)\%$, $(35.6 \pm 1.7)\%$, and $(25.8 \pm 1.8)\%$ for D_2O filtr., D_2O and H_2O samples, respectively (Fig. 4B). The D_2O sample retained the most polarization, with 38% more than the H_2O and 19% more than the D_2O filtr. sample.

Upon hyperpolarized pyruvate administration to HeLa cells, we observed production of lactate and evaluated its maximum SNR (see SI, **Section 1.5, Fig. S2**): The achieved values were 227.9 ± 9.4 , 365.3 ± 27.4 , and 156.7 ± 14.6 for D_2O filtr., D_2O , and H_2O , respectively (**Fig. 4C**). Again, the highest SNR of the metabolite was for D_2O with a 133% gain compared to H_2O and 60% compared to D_2O filtr. Statistical significance of the differences in T_1 , polarization, and SNR is indicated in the figure.

These results show that, for maximizing T_1 , the filtration step is necessary, whereas, for maximizing polarization and metabolic SNR, the transfer without radical filtration is more promising. These benefits should also translate to *in vivo* experiments, where a reasonable SNR is often even more challenging to obtain than in cell experiments. A doubling of SNR would allow reducing the flip angle (if necessary), increasing the measurement duration, or to improve measurement accuracy.

Figure 4. Effect of optimized dissolution media on hyperpolarized $[1-^{13}C]$ pyruvate and SNR of its metabolic product lactate in HeLa cell line. Three dissolution media compositions were tested: D_2O filtr. being like sample #C, D_2O like #C without the filtration step, and H_2O being like sample #G. (A) The retained polarization of pyruvate after a 45 s transfer measured at 1 T without cells was $(29.9 \pm 1.7)\%$, $(35.6 \pm 1.7)\%$, and $(25.8 \pm 1.8)\%$ for the samples as introduced. (B) The polarization lifetime of pyruvate, corrected for flip-angle and measured at 1 T without cells, was (162.1 ± 2.8) s, (153.8 ± 10.1) s, and (77.1 ± 1.2) s for the samples as introduced, respectively. (C) The observed maximum SNR for the metabolic product lactate in HeLa cells was (227.9 ± 9.4) , (365.3 ± 27.4) , and (156.7 ± 14.6) , respectively. A transfer magnet of about 35 mT was used for D_2O filtr. and D_2O , but not for H_2O . The transfer time was kept constant across all samples to accommodate about 40 s for the transfer, including filtration. D_2O and H_2O experiments were done on the same day. pH of the dissolution medium (and cells) was 7.95 ± 0.16 (7.65 ± 0.24), 7.38 ± 0.08 (7.10 ± 0.03), and 7.65 ± 0.03 (7.5 ± 0.01), respectively. Two biological repeats were performed for D_2O filtr., and three for D_2O and H_2O . Statistical significance was evaluated using one-sided t-tests with Welch-correction. Respective kinetics and analysis in SI, **Section 1.5, Fig. S2 and Tab. S2**.

Please also note the additional text in SI.

2. Additionally, the authors should show that the extended relaxation time of $1-^{13}C$ pyruvate improves performance in an in-vivo model, specifically addressing current limitations arising from the comparatively short relaxation times.

Answer: We also believe that such a study will be necessary to provide more insights into how these improvements may impact complex models, such as rodents and even humans. Luckily, all changes made don't hinder the biocompatibility, making such a translation rather straightforward. Still, we believe that such a study is outside the scope of this work, where we have now added a simpler model system (cells) as a proof of concept. We believe that this sufficiently supports our hypothesis. We cited works where it was shown that improvements

in relaxation time increase the retained polarization and, as a result, increase the method's sensitivity in vivo.

Modified text (introduction):

Tris and EDTA are commonly used in clinical dDNP experiments^{21,33} due to their biocompatibility. In addition, D₂O has recently been shown to be safe for human imaging^{32,34}. The use of D₂O prolongs T_1 relaxation times and preserves polarization levels, thereby increasing the signal-to-noise ratio (SNR) of pyruvate and its metabolite lactate in humans³².